# Abrupt structural transition in exotic molybdenum isotopes unveils an isospin-symmetric island of inversion

J. Ha[1,2,3,4], F. Recchia [2,3] ✉, S. M. Lenzi [2,3], H. Iwasaki [5,6], D. D. Dao [7], F. Nowacki [7], A. Revel [5,6], P. Aguilera [2,3], G. de Angelis [8], J. Ash[5,6], D. Bazin [5,6], M. A. Bentley [9], S. Biswas[5], S. Carollo [2,3], M. L. Cortes[8], R. Elder[5,6], R. Escudeiro [2,3,10], P. Farris [5,6], A. Gade [5,6], T. Ginter[5], M. Grinder[5,6], J. Li[5], D. R. Napoli [8], S. Noji [5], J. Pereira[5], S. Pigliapoco[2,3], A. Pompermaier [2], A. Poves[11], K. Rezynkina [2,3], A. Sanchez[5,6], R. Wadsworth [9] & D. Weisshaar[5]

Like electrons in atoms, protons and neutrons in nuclei occupy orbitals in a shell structure with energy gaps at magic numbers. Radioactive-beam experiments revealed the disappearance of magic numbers in some neutron-rich isotopes. In these nuclei, configurations involving particles excited across the shell gap gain correlation energy, becoming the ground state. Neutron-rich regions of the nuclear chart that exhibit this property are known as "Islands of Inversion". Here we present the lifetime measurement of the first $2^+$ states in $^{84}$Mo ($N = Z$) and $^{86}$Mo ($N = Z + 2$) revealing an unexpected sharp structural change between them defining the edge of the region of deformation around $^{80}$Zr. Similarly to the neutron-rich $N = 40$ Island of Inversion near $^{64}$Cr where cross-shell excitations dominate, we identify this region as an Island of Inversion with symmetrical proton and neutron excitations that we term "Isospin-Symmetric Island of Inversion". Three-nucleon forces are suggested to drive Mo isotope structural changes.

The atomic nucleus is a complex quantum many-body system characterized by regularities: "magic numbers", i.e., numbers of nucleons (proton or neutron) for which nuclei are particularly stable. This experimental evidence over the nuclear chart introduced the idea of the shell structure in nuclei. The magic numbers correspond to the complete filling of spherical orbitals, associated with large energy gaps between shell closures. Thus, nuclei with a magic number of protons and/or neutrons assume spherical shapes. Recently, thanks to the

progress in experimental instrumentation and the availability of radioactive beams, a different behavior has been observed far from the valley of stability: some magic numbers are obscured by correlations in nuclei with a large neutron excess, and collective behavior develops, associated with deformed nuclear shapes[1–7]. These regions of deformation are known as "Islands of Inversion" because the wave function configurations of the ground states of nuclei in these regions no longer correspond to the "normal" filling of the spherical orbitals. Intruder

[1]Center for Exotic Nuclear Studies, Institute for Basic Science, Daejeon, Republic of Korea. [2]Dipartimento di Fisica e Astronomia, Università Degli Studi di Padova, Padova, Italy. [3]Sezione di Padova, Istituto Nazionale di Fisica Nucleare, Padova, Italy. [4]Instituut voor Kern- en Stralingsfysica, KU Leuven, Leuven, Belgium. [5]Facility for Rare Isotope Beams, Michigan State University, East Lansing, MI, USA. [6]Department of Physics and Astronomy, Michigan State University, East Lansing, MI, USA. [7]Université de Strasbourg, CNRS, IPHC UMR7178, Strasbourg, France. [8]Laboratori Nazionali di Legnaro, Istituto Nazionale di Fisica Nucleare, Legnaro, Italy. [9]School of Physics, Engineering and Technology, University of York, York, UK. [10]Instituto de Fisica, Universidade de São Paulo, São Paulo, Brazil. [11]Departamento de Física Teórica and IFT UAM-CSIC, Universidad Autónoma de Madrid, Madrid, Spain. ✉e-mail: francesco.recchia@unipd.it

configurations involving the excitation of many nucleons across the shell gap gain binding energy, becoming dominant in the ground state. The origin of the Islands of Inversion can be explained in the framework of the shell model and dynamical symmetries, variants of the SU(3) symmetry associated with the quadrupole interaction[8]. The gain in correlation energy due to the quadrupole part of the nuclear interaction drives the nucleus towards a quadrupole-deformed shape.

While this phenomenon has been so far studied in very neutron-rich nuclei, its manifestation in nuclei residing on the proton-rich side remains to be explored in detail due to the experimental difficulty of producing medium-heavy-mass $N \sim Z$ nuclei. From the theoretical side, the increase of nuclear degrees of freedom challenges the possibility of performing microscopic shell-model calculations in these exotic regions.

Light nuclei at the $N = Z$ line present at low excitation energy deformed intruder structures arising from many particle-many hole ($npnh$) excitations across two contiguous harmonic oscillator major shells. For example, rotational bands corresponding to $4p4h$ and $8p8h$ excitations have been observed in the low-lying spectrum of the doubly magic $^{16}$O and $^{40}$Ca[9,10]. Due to the spin-orbit force, such intruder configurations may come still lower in energy and even be inverted with configurations corresponding to the normal orbital filling. Such changes of structure can be identified above $N = Z = 30$, where the nuclear shapes evolve rapidly, passing from a mild, triaxial deformation of $^{64}$Ge[11] to an oblate-prolate shape transition in $^{72}$Kr[8,12].

A very large deformation region is developed around the harmonic-oscillator magic number $N = Z = 40$, and a smooth decrease of deformation towards the spherical $^{100}$Sn is predicted[8]. The degree of nuclear deformation can be deduced from the reduced electric quadrupole ($E2$) transition probability from the first $2^+$ excited state to the ground state, B(E2; $2_1^+ \rightarrow 0_1^+$). The heaviest even-even $N = Z$ nuclei

where these probabilities have been measured so far are $^{76}$Sr and $^{80}$Zr. They show the largest reduced transition probabilities measured in $N = Z$ nuclides: B(E2; $2_1^+ \rightarrow 0_1^+$) = 2220(270)$e^2$fm[4,13] and 1910(180) $e^2$fm[4,14], respectively. Interestingly, the B(E2; $2_1^+ \rightarrow 0_1^+$) of the $N = Z + 2$ isotopes with two additional neutrons follows closely the $N = Z$ trend. Recent mean–field and Monte–Carlo shell model calculations[15,16] show enhanced collectivity in the region, but overestimate the $E2$ strength in $^{80}$Zr, while predicting a sharp decrease of the B(E2; $2_1^+ \rightarrow 0_1^+$) for the heavier $N = Z = 42$ $^{84}$Mo, with a similar value as the $N = Z + 2$ isotope $^{86}$Mo. More recently, deformed ab initio Coupled Cluster calculations have been applied to the $A \sim 80$ region. These calculations, restricted to axially symmetric shapes, do not predict large collectivity in $^{84}$Mo[17]. The difficulty of reproducing the data in this region of deformation by different models has been pointed out in recent mass measurements[18].

In this work, we present the first lifetime measurement of the $2_1^+$ states in $^{84}$Mo and $^{86}$Mo using radioactive ion beams with the state-of-the-art Gamma-Ray Energy Tracking In-beam Nuclear Array (GRETINA)[19,20] and the TRIple PLunger for EXotic beams (TRIPLEX)[21]. Contrary to previous predictions[15], the salient difference in the obtained B(E2; $2_1^+ \rightarrow 0_1^+$) between the two isotopes reveals a profound change in their structure and affords deeper insight into the evolution of the nuclear structure at the proton-rich side of the stability line.

## Results

The experiment was performed at the National Superconducting Cyclotron Laboratory (NSCL) Coupled Cyclotron Facility, Michigan State University. Figure 1 shows a schematic view of the experimental setup. The secondary beam of $^{86}$Mo was generated through the fragmentation of a primary beam of $^{92}$Mo at 140 $A$ MeV ($A$ is nucleon number), utilizing a 235-mg cm$^{-2}$ $^9$Be target positioned at the entrance of the A1900 fragment separator. Subsequently, the $^{86}$Mo secondary

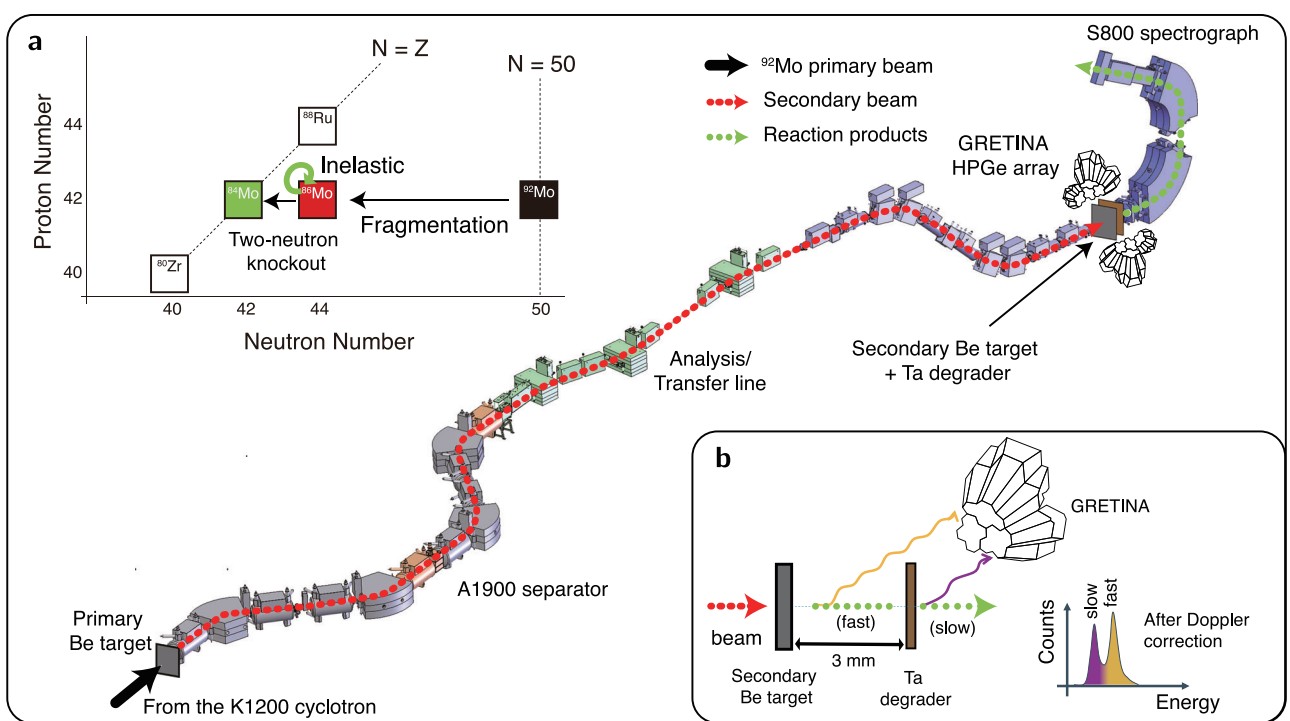

**Fig. 1 | Schematic plots of the experimental procedure. a** The secondary beam was produced by fragmentation of a $^{92}$Mo beam on a primary Be target and purified using the A1900 separator. The beam reacted with a second Be target in front of the GRETINA array. The reaction products were identified using the S800 spectrograph. **b** Principle of the lifetime measurement using the TRIPLEX plunger. The excited reaction products can experience γ-decay between the target

and a Ta degrader foil, at high velocity (v/c=0.365 for $^{86}$Mo), or after the degrader, at a lower velocity (v/c=0.324 for $^{86}$Mo), producing two peaks in the Doppler corrected γ-ray spectrum. The observation of the ratio between the statistics in the slow (purple) and fast (yellow) components provides information on the lifetime of the excited state.

beam, averaging an energy of 103 $A$ MeV, was selected by the separator, ensuring a 1% momentum acceptance. The intensity of the secondary beam reached approximately $2 \times 10^5$ particles per second in total, with a purity of 3.2% specifically for $^{86}$Mo.

Excited states in $^{86}$Mo and $^{84}$Mo were populated through inelastic and two-neutron removal reactions on a $^9$Be secondary target, respectively. Gamma rays were detected by GRETINA, which consisted of twelve quad modules. Each quad module is composed of four n-type, 36-fold segmented high-purity Ge crystals[20].

The signals are digitized and decomposed[19] to obtain the γ-ray interaction points. This is crucial for the Doppler-shift correction as the γ rays are emitted from nuclei recoiling at a large velocity. In this experiment, four quads were placed in the 58° ring and eight in the 90° ring with respect to the GRETINA center on the beam axis. Gamma-ray energies were reconstructed by summing the energy seen by neighboring crystals using the add-back method described in ref. 22.

The TRIPLEX plunger device was employed with a 1.325-mm-thick $^9$Be secondary target and a 101-μm-thick $^{181}$Ta degrader separated by a distance of 3 mm, as depicted in Fig. 1b. A 7-mg cm$^{-2}$-thick polyethylene foil was mounted behind the Ta degrader to increase the fraction of ions in the fully stripped charge state. The target in the TRIPLEX was mounted 202 mm upstream of the center of GRETINA to obtain a larger efficiency at forward angles, thereby optimizing the GRETINA array for the lifetime measurement. The reaction products were identified by time-of-flight and energy loss in the S800 spectrograph[23]. The momentum of the ejectile from the reaction target, reconstructed from the position and angle at the focal plane of the S800 spectrograph, was used for the Doppler correction on an event-by-event basis.

Figure 2 shows the γ-ray spectra obtained from the two-neutron removal and from the inelastic excitation reaction channels. The positions of the transitions $2_1^+ \rightarrow 0_1^+$ and $4_1^+ \rightarrow 2_1^+$ are marked in each spectrum. The Doppler correction was performed using the velocity of the ions upon exiting the degrader as determined from the magnetic rigidity in the S800 spectrograph. Gamma rays emitted at such velocity generate the slow component of the peaks (S), while γ rays emitted before the degrader give rise to the fast components (F).

In Fig. 2b it is possible to observe only the fast component of the transition $4_1^+ \rightarrow 2_1^+$ in $^{86}$Mo, which implies a short lifetime for the first $4^+$ state and hence a limited impact on the first $2^+$ lifetime estimation, as included in the evaluation of systematic errors described below.

The experimental spectra were reproduced using the GEANT4 simulation package[24] that incorporates the geometry of the experimental setup[12,13,25]. The spatial and energy distributions of the secondary beam were reproduced in the simulation.

The population of higher excited states was evaluated from the γ-ray intensities. For $^{86}$Mo, 50(10)% of the population to the first $2^+$ state originates from feeding of the first $4^+$ state. We have taken into account the feeding from the first $4^+$ state in $^{86}$Mo, according to a least-$\chi^2$ fitting for the $^9$Be($^{86}$Mo, $^{86}$Moγ) spectrum shown in Fig. 2b. The estimation could not be performed for $^{84}$Mo because of the limited statistics. Instead, we assumed that the feeding ratios to the first $4^+$ state in $^{84}$Mo and $^{86}$Mo were the same, but with larger uncertainty [50(20)%] in $^{84}$Mo.

The fitting procedure followed the prescriptions described by Adrich and collaborators[25]. This consists of a least-$\chi^2$ method where both the lifetime of the first $2^+$ state and the peak height of the $2_1^+ \rightarrow 0_1^+$ transition are varied. To reduce the degrees of freedom, we froze the background fitting parameters. Moreover, only the bins with a significant number of counts with respect to the background level have been considered to maximize the sensitivity.

From the comparison with the simulation, the lifetimes of the first $2^+$ state in $^{86}$Mo and $^{84}$Mo were determined to be $19.9^{+1.9}_{-1.7}$(stat.)$^{+0.7}_{-1.1}$(syst.) ps and $27.1^{+8.8}_{-6.7}$(stat.)$^{+0.8}_{-0.9}$(syst.) ps, respectively. The finite lifetime of the $4^+$ states, as well as the degrader-to-target yield ratio, contribute to the systematic uncertainty of the lifetime of the $2^+$ state. Such a contribution is estimated based on the previous measurement[26].

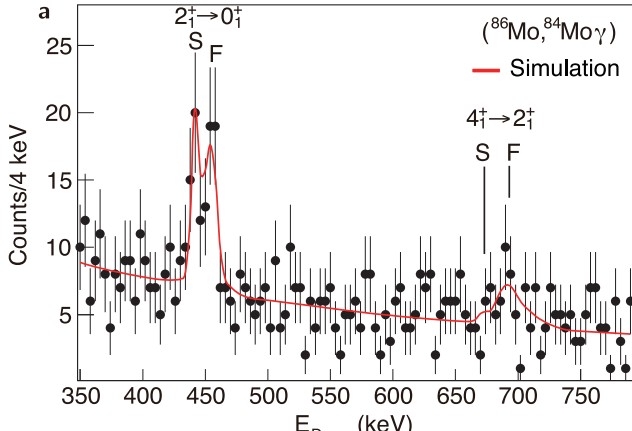

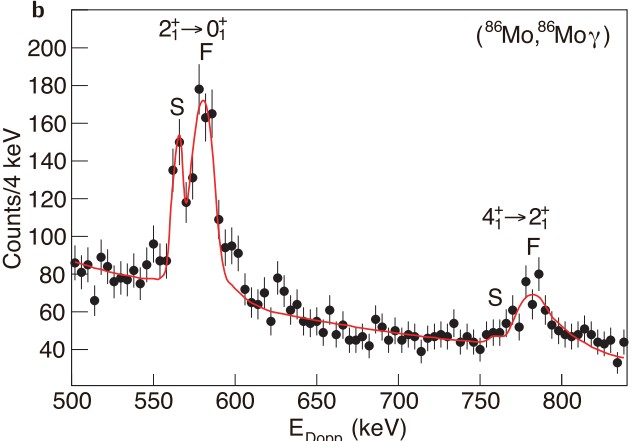

**Fig. 2 | Comparison of gamma-ray spectra with simulation. a** Gamma-ray spectrum observed in coincidence with the two-neutron knockout reaction $^9$Be($^{86}$Mo, $^{84}$Moγ). A simulated spectrum with $\tau(2_1^+) = 27.1$ ps (red line) is drawn for comparison. The energies of the $2_1^+ \rightarrow 0_1^+$ and $4_1^+ \rightarrow 2_1^+$ transitions are 443.9 keV and 673.4 keV[50], respectively. **b** Gamma-ray spectrum observed in coincidence with the inelastic excitation of $^{86}$Mo. A simulated spectrum with $\tau(2_1^+)$=19.9 ps (red line) is drawn for comparison. The energies of the $2_1^+ \rightarrow 0_1^+$ and $4_1^+ \rightarrow 2_1^+$ transitions are 566.6 keV and 760.9 keV[51], respectively. The error bars show the 1σ statistical uncertainty. The fast and slow components of the Doppler corrected photopeaks are labeled "F" and "S", respectively.

## Discussion

The lifetimes measured in the present work yield reduced transition probabilities B(E2; $2_1^+ \rightarrow 0_1^+$) = $1740^{+580}_{-430}e^2$fm$^4$ for $^{84}$Mo and 707(71) $e^2$fm$^4$ for $^{86}$Mo. They are reported in Fig. 3 together with the B(E2; $2_1^+ \rightarrow 0_1^+$) for $N=Z$ and $N=Z+2$ nuclei, from Ge to Mo isotopes. At odds with the systematic trend of lighter nuclei, where the B(E2; $2_1^+ \rightarrow 0_1^+$) for all isotopic pairs remain closely aligned, a remarkable reduction of the transition probability is observed for $^{86}$Mo, while $^{84}$Mo maintains a large B(E2; $2_1^+ \rightarrow 0_1^+$).

Between the doubly magic $^{56}$Ni and $^{100}$Sn, the shells lying at the Fermi surface define a natural valence space composed of the $2p_{3/2}$, $1f_{5/2}$, $2p_{1/2}$, and $1g_{9/2}$ orbitals for both protons and neutrons[27]. To account for the rapid change of collectivity around mass $A \sim 80$ along the $N \sim Z$ line, we extend the aforementioned valence space with the addition of the two extra orbitals $2d_{5/2}$ and $3s_{1/2}$, which are the minimum of the above valence space to account for the quadrupole degrees of freedom of the system[8]. The resulting valence space, in the context of the shell model, contains the upper three orbits of the $pf$ shell and the lower three orbits of the $sdg$ shell. In the framework of the dynamical symmetries, variants of SU(3), these two orbital blocks correspond to the pseudo-SU(3) and quasi-SU(3) symmetries, respectively. Their quadrupole properties have been

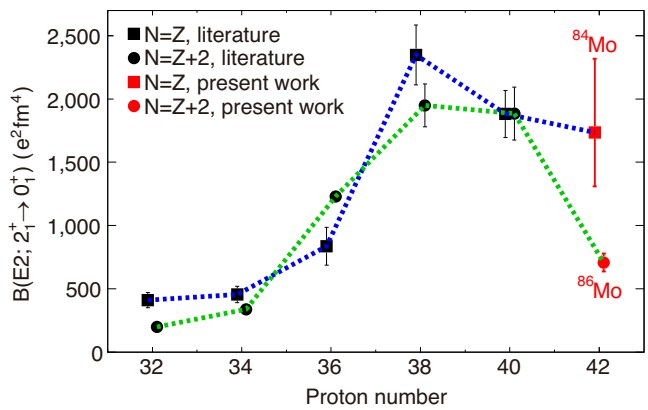

**Fig. 3 | Systematics of the B(E2; $2_1^+ \rightarrow 0_1^+$) along the $N = Z$ and $N = Z + 2$ isotopes.** The statistical and systematic uncertainties were taken into account for $^{84}$Mo and $^{86}$Mo. The B(E2; $2_1^+ \rightarrow 0_1^+$) were estimated to be $1740^{+580}_{-430}$ $e^2$fm$^4$ for $^{84}$Mo and 707(71) $e^2$fm$^4$ for $^{86}$Mo. Previous data are taken from refs. [11–14,52–59].

**Table 1 | Computed B(E2; $2_1^+ \rightarrow 0_1^+$) values**

| nuclide | np-nh | B(E2; $2_1^+ \rightarrow 0_1^+$) ($e^2$fm$^4$) | | | | |
|---|---|---|---|---|---|---|
| | | QPSU3 | PHF | DNO-SM | SM | Exp |
| $^{84}$Mo | 4p-4h | 1104 | 1121 | 1512 | - | $1740^{+580}_{-430}$ |
| | 8p-8h | 1891 | 1794 | | | |
| $^{86}$Mo | 0p-0h | 424 | 196 | 893 | 731 | 707(71) |
| | 2p-2h | 1143 | 871 | | | |
| | 4p-4h | 1416 | 1179 | | | |
| | 6p-6h | 1858 | 1655 | | | |

The values are obtained using different models: quasi/pseudo-SU(3) estimates (QPSU3) and Projected Hartree-Fock solutions (PHF) for different np-nh configurations, DNO shell-model calculations (DNO-SM), shell-model diagonalization (SM), and experimental values (Exp) for $^{84,86}$Mo.

analyzed in detail in ref. 8. We note that the remaining $1g_{7/2}$, $2d_{3/2}$, and $1h_{11/2}$ shells are omitted for two reasons: firstly, nuclear systems are known to exhibit evolving shell structure with the proton and neutron numbers, and these orbitals appear to lie high in energy in $^{91}$Zr; secondly, they affect the quadrupole coherence only at a perturbative level. For the quasi +pseudo-SU(3) valence space, we have derived a new effective interaction based on realistic matrix elements with monopole constraints to emulate three-body forces, named hereafter DNP-ZBM3. The effective charges used in the present valence space are 1.5 and 0.5 for protons and neutrons, respectively, as derived and discussed in ref. 8.

In the case of $^{84}$Mo and $^{86}$Mo, it is possible to assess the collectivity with different levels of approximation. Under the assumption of a configuration based on a fixed number of particle-hole excitations across $N = Z = 40$, we use two different methods:

- QPSU3—a pure quadrupole force within the quasi- and pseudo-SU(3) dynamical symmetries, variants of SU(3)[8];
- PHF—a single angular-momentum projected deformed Hartree-Fock state[28,29];

It is possible to exploit a diagonalization within the shell model:

- DNO-SM—with a mixing of projected deformed Hartree-Fock states and variation-after-projection (VAP), using the recently developed Discrete Non-Orthogonal Shell Model[29,30];
- SM—with the exact shell-model calculation[31].

The last three calculations have been performed with the full DNP-ZBM3 Hamiltonian. A detailed description of these methods is given in the section "Methods". Here, we summarize the results obtained with the different approaches.

When a pure quadrupole force is considered (QPSU3), it is possible to calculate the B(E2; $2_1^+ \rightarrow 0_1^+$) following the heuristic algebraic approach described in ref. 8 and applied in refs. 32,33. The results obtained for different np-nh configurations are reported in the QPSU3 column of Table 1. The large B(E2; $2_1^+ \rightarrow 0_1^+$) value that we experimentally measured for $^{84}$Mo compares well with the 8p-8h configuration. In contrast, the lower B(E2; $2_1^+ \rightarrow 0_1^+$) observed in $^{86}$Mo suggests a rather different structure. It is important to keep in mind that the QPSU3 values represent the upper limit of maximum deformation, and hence collectivity, allowed by the model space.

When considering the angular-momentum projection of a single deformed HF state at a given np-nh structure, it can be seen that these results, reported in the PHF column 4 of Table 1, are in very good agreement with the QPSU3 estimates. Similarly, the comparison with the experimental data suggests that the 8p-8h configuration is favored

for $^{84}$Mo, while a mixing of different np-nh configurations may be expected for $^{86}$Mo.

In the DNO-SM diagonalization[29,30], in contrast to the PHF case, where only one deformed HF state is considered, we first construct the full landscape of the potential energy surface (PES). Here, each deformed Hartree-Fock basis state is a single ($\beta$, $\gamma$) point, where $\beta$ is the quadrupole deformation parameter, whereas $\gamma$ measures the deviation from axial symmetry. Hence, the PES landscape provides us with the possibility of looking at the relevant deformations that may compete. A diagonalization over this subset of basis vectors is used to obtain the amplitude of each component. Figure 4 shows, with orange circles, the amplitudes of each ($\beta$,$\gamma$) basis state present in the ground state of both isotopes.

A remarkable change in structure between the two isotopes is apparent from the figure. On one hand, a very complex mixing of configurations across the PES is observed for the wave function of $^{86}$Mo. On the other hand, $^{84}$Mo develops around a largely deformed triaxial shape, and its ground-state wave function is dominated by fewer components, all of 8p-8h nature. (see "Methods" subsection "Shell Model and Discrete non-orthogonal shell model") This result clearly indicates a striking transition occurring in these highly deformed $N \sim Z$ nuclei. After the diagonalization of the DNP-ZBM3 Hamiltonian, we obtain the B(E2) values reported in column 5 of Table 1, in very good agreement with the experimental findings.

Finally, we perform the SM calculation of the B(E2) for $^{86}$Mo and report it in Table 1. It reproduces the data with excellent accuracy. The dimension of this calculation ($10^{11}$ basis states) is among the largest ever achieved for an exact diagonalization with the present computational capabilities. For $^{84}$Mo the exact diagonalization exceeds this limit.

The above analysis, pointing to shape and structure transitions between $^{84,86}$Mo, calls for an explanation. From the shell-model perspective, the reason for this transition is likely related to important shell evolution features. On one side, the size of $N = Z = 40$ shell gap must be small enough to favor intruder configurations that drive deformation into the system. On the other side, when starting to add neutrons between $N = 40$ and $N = 50$, the $g_{9/2}$–$d_{5/2}$ shell gap should increase to hinder quadrupole correlations and recover magicity in $^{92}$Mo. Indeed, the very nature of shell evolution, as pointed out in early studies by Zuker and collaborators[31,34,35] is driven by the properties of the monopole part of the shell-model Hamiltonian. Such properties have been recently shown to be strongly correlated to three-nucleon (3$N$) forces[36], suggesting that they are the key behind the structure transition we observed in $^{84,86}$Mo. The monopole part of the 3$N$ force induces a reduction of the $N = Z = 40$ harmonic-oscillator shell gap while restoring the spin-orbit $g_{9/2}$ shell closure, in agreement with the experimentally observed shell structure as depicted in Fig. 5. In other words, we witness here a remarkable case where 3$N$ interactions are at work in deformed systems. So far, the role of 3$N$ forces has been shown

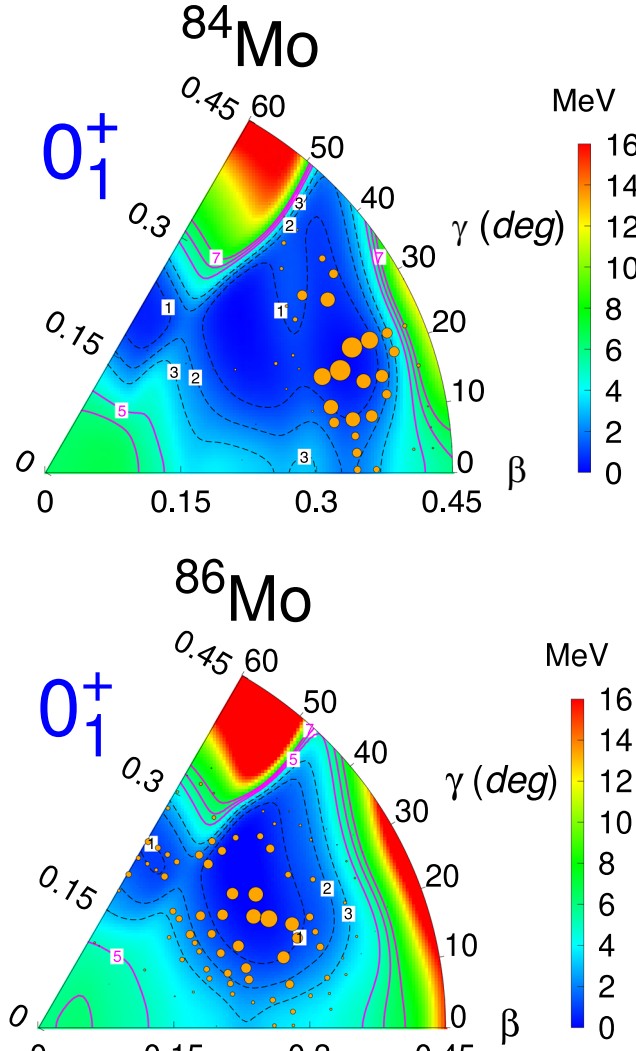

**Fig. 4 | Potential energy surfaces for $^{84,86}$Mo.** The surfaces are obtained with the DNP-ZBM3 effective interaction. The area of the orange circles is directly proportional to the probability of the configuration with ($\beta, \gamma$) deformation parameters in the ground state wave function.

to be crucial to reproduce magicity at the spin-orbit closures in spherical nuclei [34,36–39]. However, in deformed systems, the impact of 3$N$ versus two-nucleon (2$N$) forces is more difficult to put in evidence due to the strong mixing of orbitals. In the present $A = 80$ region, this 3$N$ impact should be enhanced, producing intruder effects as we explained above.

To clarify this, we have computed the B(E2) values in $^{84,86}$Mo, with the use of a V$_{lowk}$-softened N$^3$LO 2$N$ interaction [40]. While these calculations are only illustrative and cannot be a full proof for a genuine 3$N$ force effects, they put in evidence that, in the absence of 3$N$ forces, no intruder effects are obtained. Moreover, they underestimate the $E2$ strength by more than one order of magnitude for both nuclei, producing a B(E2) of 174 $e^2$fm$^4$ for $^{84}$Mo and 0.63 $e^2$fm$^4$ for $^{86}$Mo. This is due to the ill-defined shell gaps(the $N = Z = 40$ gap is overestimated by several MeV) and shell ordering (the spin-orbit hierarchy is not respected and low-$\ell$ orbits are more bound than large-$\ell$ ones) as shown in Fig. 5.Such flaws are removed in our DNP-ZBM3 Hamiltonian with phenomenological monopole corrections, emulating the 3$N$ forces behavior.

From the above discussion, we conclude that the structural transition between $^{84}$Mo and $^{86}$Mo defines the border that closes the highly deformed region around $A \sim 80$. So far, several instances of Islands of Inversion have been observed, but only in the neutron-rich part of the nuclear chart, at the magic numbers $N = 8, 20, 28, 40$, as depicted in Fig. 6. It is well known that at the proton-rich side, along the $N = Z$ line, around mass $A = 80$, a highly deformed region develops. The comparison of the experimental data with theory allows us to interpret the high degree of deformation in this region as due to the *coherent action* of protons and neutrons with up to 4 proton and 4 neutron particle-hole excitations across the $N = Z = 40$ shell gap. Such a situation is very similar to the one encountered for the neutron-rich $^{64}$Cr at $N = 40$, where the intruder configuration is described by a 4-neutron particle-hole excitation. The present study thus reveals an interesting isospin-symmetric situation, giving rise to what we call here the Isospin-Symmetric Island of Inversion.

In summary, the lifetimes of the first 2$^+$ states in $^{84}$Mo and $^{86}$Mo have been measured using the plunger setup coupled to GRETINA. The extracted reduced transition probabilities, B(E2; $2_1^+ \rightarrow 0_1^+$), point to a large difference between the $N = Z$ and $N = Z + 2$ Mo isotopes. State-of-the-art calculations, using the interacting shell model and the DNO-SM approaches, are in very good agreement with the measured values. These calculations nicely explain the sudden change of deformation observed between the two isotopes as originating from the increase of

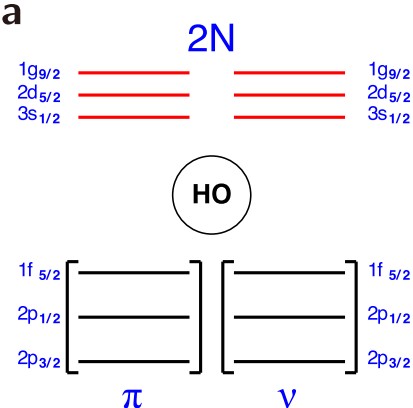

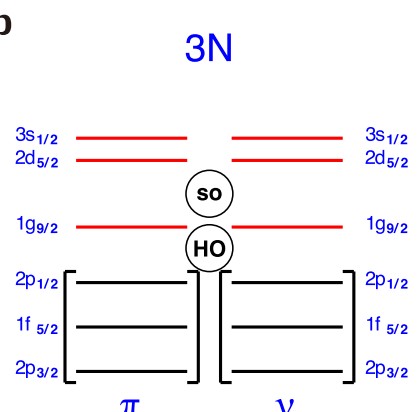

**Fig. 5 | Schematic shell evolution.** Schematic shell evolution without (**a**) and with (**b**) 3N forces: from left to right, the Harmonic Oscillator (HO) $N = 40$ shell gap $p_{1/2} - g_{9/2}$ is decreased (as described in ref. 35), and the spin-orbit (so) $N = 50$ shell gap is restored, and the correct shell order is reestablished. (Similar to what is shown in Fig. 7 of ref. 38).

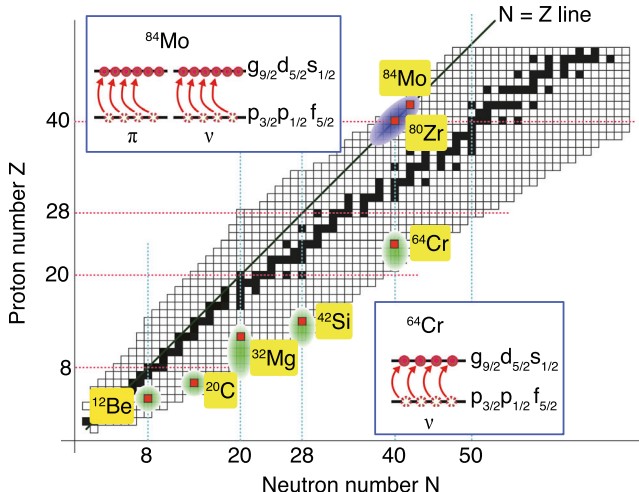

**Fig. 6 | Schematic view of Islands of Inversion.** The Islands of Inversion are marked in green, along the nuclear chart. The bottom inset shows a 4p-4h configuration in neutron orbitals leading to the Island of Inversion in $^{64}$Cr. The blue region along $N=Z$ line shows the Isospin-Symmetric Island of Inversion. The 8p-8h excitations in $^{84}$Mo are shown in the upper inset.

the neutron $g_{9/2}$–$d_{5/2}$ shell gap brought by the two extra neutrons, a fingerprint of 3$N$ forces.

Our analysis clearly points to a sharp structural transition which places $^{84}$Mo inside and $^{86}$Mo at the edge of a newly defined Island of Inversion in the $A \sim 80$ region. The strong collectivity observed in $^{84}$Mo is understood as the coherent quadrupole $T = 0$ excitations developing between two major Harmonic Oscillator shells on both proton and neutron sides, as already pointed out in ref. 15. Figure 6 shows the different Islands of Inversion in the nuclear chart. In the neutron-rich region, they are found at the magic numbers $N = 8, 20, 28, 40$. The present experimental findings place the limits of such a unique Isospin-Symmetric Island of Inversion that can be found in the proton-rich side, interpreted as produced by simultaneous proton and neutron intruder configurations.

## Methods
### Algebraic SU(3) analysis
From the pioneering work of Elliot[41,42], in the framework of the SU(3) symmetry, in a main shell, with degenerate orbits and in the presence of a pure quadrupole interaction, the configuration that maximizes the quadrupole moment becomes the ground state. However, the spin-orbit interaction breaks the degeneracy of the orbits and destroys the SU(3) symmetry; its variants, pseudo-SU(3)[43] and quasi-SU(3)[44] apply in reduced shell-model spaces[8]. We recall here that the pseudo-SU(3) dynamical symmetry is realized in a harmonic oscillator shell when the lowest large $j$ shell is closed. Instead, the quasi-SU(3) dynamical symmetry is valid when $\Delta j = \Delta \ell = 2$ orbits are present at the Fermi surface. In the proton-rich $A = 80$ region, the pseudo-SU(3) symmetry applies in the space spanned by the $2p_{3/2}$, $2p_{1/2}$, and $1f_{5/2}$ orbits, while the quasi-SU(3) symmetry applies in the space formed by the $1g_{9/2}$, $2d_{5/2}$, and $3s_{1/2}$ orbits. The combination of these valence spaces forms the ZBM3 model space, in analogy with the ZBM[45] and ZBM2[46] valence spaces, around $N = Z = 8$ and $N = Z = 20$, respectively.

By diagonalizing the quadrupole force in the ZBM3 space, the single-particle quadrupole moments are obtained. Normalized quadrupole moments, $Q_0/b^2$, with $b$ being the harmonic oscillator size parameter, are plotted in Supplementary Fig. 1. The quadrupole moment of the nucleus is obtained by summing up the single-particle moments. In the case of $^{84}$Mo, the maximum quadrupole moment built

up by the 14 protons and 14 neutrons above the core of $^{56}$Ni is obtained by filling the Nilsson-SU(3) "orbits" as shown in Supplementary Fig. 1, corresponding to 12 particles in the quasi-$sdg$ orbits and 8 holes in the pseudo-$pf$ orbits. This implies that particle-hole excitations from the $fp$ to the $gds$ spaces play a relevant role. The reduced transition probabilities resulting from these quadrupole moments[8] are reported in the third column of Table 1, for different particle-hole configurations. From the obtained values, we can infer that $^{84}$Mo should be of 8$p$-8$h$ intruder nature, while $^{86}$Mo appears to have a less pronounced intruder nature.

### Shell model and discrete non-orthogonal shell model
The newly constructed DNP-ZBM3 effective interaction includes matrix elements from the JUN45 interaction[27], complemented with matrix elements of the LNPS interaction[5], which originate from the Kahana-Lee-Scott realistic interaction[47]. Ongoing study for a global description of this region using this interaction will be presented elsewhere. Shell-model (SM) diagonalization using the ANTOINE code[31,48] has been performed up to 10$p$-10$h$ excitations across the $N = Z = 40$ shell gap for $^{86}$Mo. This calculation, with a typical dimension of about ~$10^{11}$, is the largest that can be afforded by our present numerical capacity for exact diagonalizations. The SM diagonalization in the spherical $M$-scheme harmonic oscillator basis starts with the lowest excitations 0$p$-0$h$, 2$p$-2$h$, …and runs up to 10$p$-10$h$.

In contrast, the DNO-SM calculation uses a deformed Hartree-Fock (HF) basis where each basis state can be represented by a single point ($\beta$, $\gamma$) in the Potential Energy Surface (see Fig. 4). The full wave function is thus a linear combination of many deformed HF states. In order to select relevant HF states in the potential energy surface, we proceed from the highly deformed Hartree-Fock minimum of 8$p$-8$h$ for $^{84}$Mo and 4$p$-4$h$ for $^{86}$Mo. The additional correlations are then added by the Caurier minimization technique using the CARINA code, as exposed in ref. 29, where the wave function expansion is iteratively optimized through the minimization of the ground-state energy. As pointed out by our analysis, in the case of $^{86}$Mo, the resulting DNO-SM calculation as such shows a considerable amount of mixing across multiple particle-hole configurations in the potential energy surface (see Supplementary Table 1 and Fig. 4). A heuristic explanation is explored in Supplementary Fig. 2 where the variation of the Hartree-Fock energy with respect to the quadrupole deformation parameter $\beta$ is plotted at fixed $\gamma$ corresponding to the HF minimum. The deformation curve in Supplementary Fig. 2 shows two distinct 4$p$-4$h$ and 8$p$-8$h$ minima in $^{84}$Mo with a high "energy barrier" separating them, whereas for $^{86}$Mo, there is less of a clear sign of an 8$p$-8$h$ minimum. This may be the reason that favors the configuration mixing in one case and prevents it from happening in the other. Consequently, the distribution of the configurations of $^{84}$Mo and $^{86}$Mo across the potential energy surfaces differs significantly, as depicted in Fig. 4.

To better assess the B(E2) transition probabilities convergence, the DNP-ZBM3 effective Hamiltonian is further diagonalized within the Variation After Projection extension of the DNO-SM, recently developed in[30]. This approach consists of a double variation of both the mixing amplitudes and the intrinsic states simultaneously, conserving the symmetries of the Hamiltonian. For example, the rotational and parity symmetries in our case are preserved at the same time by means of projection techniques[49]. Physically, as shown in ref. 30, this method is equivalent to the incorporation of multiple particle-hole excitations on top of spherical or deformed reference Slater states. Hence, it amounts to capturing correlations brought by these particle-hole excitations through the configuration mixing of intricate symmetry-breaking Slater states. In Table 1, we report the corresponding obtained B(E2) values for $^{84,86}$Mo in comparison with the SM diagonalization and the experimental data.

## Data availability
Figures 2, 3, 4, and Supplementary Fig. 2 are derived from the source data in the Supplementary information/Source Data file. The rest of the experimental data used to support the findings of this study are available from the principal investigator (corresponding author) upon request. Source data are provided with this paper.

## Code availability
Technical methods used in our unpublished computer codes to produce the results in this paper have been detailed in prior published works. Requests for further explanations of these computational techniques are available from the corresponding author.

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

## Acknowledgements

The authors thank the NSCL operation department for their efforts in the success of this experiment. A.O. Macchiavelli and E. Clement are acknowledged for fruitful discussions. J.H. and F.R. acknowledge the support of the Department of Physics and Astronomy of the University of Padova, project "BIRD185448". D.D.D. and F.N. acknowledge the support of the International Research Laboratory for Nuclear Physics and Astrophysics (IRL NPA) CNRS laboratory, the hospitality of the NSCL-FRIB laboratory, and the financial support from CNRS/IN2P3, France, via ABI-CONFI Master project. D.D.D. acknowledges the financial support from CNRS/IN2P3, France, through the budget for an early-career CNRS researcher. This work was supported by the U.S. Department of Energy (DOE), Office of Science, Office of Nuclear Physics, under awards nos. DE-SC0023633 and DE-SC0020451, by the U.S. National Science Foundation (NSF) under grant no. PHY1565546 (Operation of the NSCL), and by the DOE National Nuclear Security Administration through the Nuclear Science and Security Consortium, under award no. DE-NA0003180. GRETINA was funded by the DOE, Office of Science. Operation of GRETINA at NSCL was supported by DOE under grants nos. DE-SC0019034 (NSCL) and DE-AC02-05CH11231 (LBNL). This work was supported by the Institute for Basic Science (IBS-R031-Y3); IRI-I002619N of the FWO Research Foundation—Flanders. A.P. acknowledges support of the grant nos. CEX2020-001007-S funded by MCIN/AEI/10.13039/501100011033 and PID2021-127890NB-I00.

## Author contributions

F.R. and S.M.L. proposed and led the experiment. H.I. coordinated the preparation of the experiment. H.I., A.R., J.A., D.B., S.B., R. Elder, P.F., A.G., T.G., M.G., J.L., S.N., J.P., A.S., and D.W. performed the experiment. J.H., F.R., S.M.L., M.L.C., D.R.N., and R.W. checked the online data taking. J.H. performed the data analysis. F.R. supervised the data analysis. P.A. and S.C. made special contributions to the data analysis. J.H. performed the Monte–Carlo simulations for the lifetime measurement with the contribution of A. Pompermaier, J.H., F.R., S.M.L., H.I., P.A., G.de A., M.A.B., S.C., M.L.C., R. Escudeiro, S.P., and K.R. contributed and discussed the data analysis, the Monte–Carlo simulations, and the experimental results. D.D.D. and F.N. performed the theoretical calculations and interpreted the results, A.P. discussed them. J.H., F.R., S.M.L., D.D.D., and F.N. prepared the manuscript. All authors contributed to the manuscript.

## Competing interests

The authors declare no competing interests.
