## [Transparent Peer Review file · Nature Communications]

Abrupt Structural Transition in Exotic Molybdenum Isotopes unveils an Isospin-Symmetric Island of Inversion

Corresponding Author: Professor Francesco Recchia

Version 0:

Reviewer comments:

Reviewer #1

(Remarks to the Author)

The authors addressed satisfactorily most of my comments. I find this work interesting and important enough to be published in Nature Communications.

Prior to recommending the publication, I ask the authors to further refine their discussion of the results obtained with the Vlow-k N3LO 2N interaction. An argument is made that the origin of the structural change between the two Mo isotopes are due the three-nucleon forces that drive the properties of the monopole part of the shell-model Hamiltonian. It is clear from Ref. [34], where the importance of 3N forces for the monopole part of the Hamiltonian was revealed, that effect comes from the genuine 3N force, i.e., the chiral 3N force or the force that originates from the Delta or other resonance excitations of nucleons. The discussed calculations with the Vlowk-softened N3LO two-nucleon interaction neglect not only the genuine 3N force but also the induced 3N interaction due to the Vlowk softening and due to the valence space truncation. I appreciate that the authors attempted a VS-IMSRG calculation and recognize that those are challenging in the present context. However, at the minimum, it should be acknowledged that the Vlowk N3LO 2N calculations are illustrative only and do not prove a genuine 3N force effect.

Reviewer #2

(Remarks to the Author)

Dear Authors,

Authors took about one year to prepare responses to the reviewer's comments and to revise the manuscript. Although I acknowledge the considerable effort you have made during this period, I must state that this manuscript is not suitable for Nature Communications. The reasons are as follows:

1. Title of this manuscript, 'Isospin-Symmetric Island of Inversion' is not fit to the contents of your manuscript. If you wish to describe this mass region as the "Island of Inversion", you need to include a discussion of the nuclear structure of Sr, Zr, and Mo isotopes. If you intend to prepare a separate theoretical manuscript addressing this point, that version could be submitted to Nature Communications.
2. Your experimental results for Mo-84 and Mo-86 are very important for nuclear physics. Therefore, I recommend that you submit this draft to another journal, such as Physical Review C, Physics Letters B, or a similar venue, as soon as possible, since one year has already passed since the initial submission.

Reviewer #3

(Remarks to the Author)

Referee report on revised Isospin-Symmetric island of Inversion by J. Ha et al.

As noted in my report on the previous version of this paper, it reports on lifetime measurements of the first-excited states of the neutron deficient isotopes ^{84}Mo and ^{86}Mo , at and near $N=Z$, respectively. These lifetime measurements give electric quadrupole ($E2$) transition strengths that indicate the degree of collectivity in the $2^+ \rightarrow 0^+$ transition. High values of the $E2$ transition strength indicate deformed nuclei. Here, the observed $E2$ strength is high for ^{84}Mo ($N=Z$), similar to the adjacent $N=Z$ isotopes of Sr and Zr, whereas the transition strength is reduced for ^{86}Mo in contrast to the behavior of the $N=Z+2$

isotopes of Sr and Zr. The measurements were performed on radioactive beams and using the GRETINA gamma-ray detector array coupled to the S800 spectrograph at NSCL. They represent experiments at the forefront of the field.

The paper then follows with a detailed theoretical analysis, again at the forefront of the field, and pushing the computational limits of the nuclear shell model. It is argued that the strong E2 transition in ^{84}Mo is due to 8-particle-8-hole excitations across the N=40 gap, which does not occur for ^{86}Mo . The experimental results along with the theoretical calculations are interpreted as evidence of an "island of inversion" around N=40, which is of interest because to date "islands of inversion" have been associated with neutron-rich nuclei.

I believe that this work is worthy of publication in Nature Communications. The authors have responded appropriately to my previous report. However, in reading the revised manuscript, now some 12 months later, I have a few more comments for consideration and some minor corrections to note.

My first concern is one of terminology, in line with my first remark previously about the term "Island of Inversion". The authors have already made changes in line with my previous remarks, but I find that in the last 12 months I have become more sensitive to statements like "magic numbers disappear", and what the phrase "evolution of shell structure" really means. These terms are used in lines 10 and 11 of the Introduction. In Islands of Inversion the shell structure does not really disappear: it is still there but it is obscured by correlations that lower the energy of many-particle many-hole deformed configurations that dominate in the ground and low-lying states. The authors take this view in the remainder of the Introduction and indeed it is a theme of the paper. From this perspective it is more accurate to say "magic numbers seem to disappear".

I also recommend consideration of the phrase "evolution of shell structure" (line 11). As I see it, "evolution of nuclear structure" has a clear meaning in terms of what is observed. "Evolution of shell structure", however, can be confusing because it must be inferred indirectly from what is observed and it depends on both single-particle energies and correlations, which are interrelated, as is evident in the body of this paper. In operational terms, the single particle energies depend on the basis space and on the interactions, which must be tuned to the basis space. An Island of Inversion is a region where the basis space must be opened up to include highly correlated np-nh excitations across the shell gap, which appear at lower energy than those in the normal (single major shell) basis space. This view accords with the theme of the paper. It is not obvious that this is a "consequence of the evolution of shell structure". It is rather shell structure plus correlations.

I strongly recommend revising the use of these phrases in lines 10 and 11 of the Introduction or otherwise making it explicit what the authors mean by them, particularly for a journal with an intended broader audience.

Now to some more specific comments and suggestions:

p. 4 end of 1st paragraph. Is the 2 e5 particles per second for the ^{86}Mo or is this the total rate for all isotopes in the secondary beam? I guess the ^{86}Mo rate but please clarify.

p.4 in Figure 1 caption. I suggest giving the actual v/c, say for the ^{86}Mo scattering case as an example, rather than just stating "higher" and "lower" velocity.

p. 5. Fig. 2b. Can you comment on what the features might be at about 590 keV and 680 keV in the gamma-ray spectrum, which seem to be above background? What could be the impact of the feature at 590 keV be on the fast-peak intensity, and to what extent might it affect the deduced lifetime?

p. 6. Last line of first paragraph regarding the impact of the short 4+ lifetime could perhaps be better phrased something along the following lines: "... which implies a short lifetime for the first 4+ state and hence a limited impact on the 2+ lifetime estimation, as included in the evaluation of systematic errors described below."

p. 6. Last line of Results section. This sentence, which has been changed in the revision process, is hard to follow. I don't know what is meant. I thought the previous wording was clear (but maybe it was not accurate?). Please amend.

p. 8. Please add references to the various theoretical methods listed, specifically for QPSU3 and PHF. DNO-SM has references already given. The SM is probably well enough known that it doesn't require a reference. The PHF, however, appears to be a little unusual in that it considers a deformed HF state of specified np-nh structure, projects out 0+ and 2+ states and thus evaluates the B(E2). Is there a reference with detail on the procedure used? This would be useful for those who wish to understand a little more of the technicality behind these calculations.

Some minor corrections:

p. 2. Line 6 of Introduction. Insert "a" in "nuclei with a magic number..."

p. 5. Fig. 2a and 2b captions. "The energy of the ... transitions are..." should be "The energies of the ... transitions are..."

p. 6. 2nd last line of 3rd paragraph. Should be "feeding ratios" not "feeding ratio".

p. 6. 4th paragraph line 2. "consists of a least" rather than "consists on a least"

p. 6. 5th paragraph line 1. "lifetimes of the first ... were determined" rather than "lifetime of the first ... were determined"

p. 7. About the middle of the text. Perhaps add “symmetries” to read “correspond to pseudo-SU(3) and quasi-SU(3) symmetries, respectively.”

p. 9. 2 lines under Table 1. The full stop should be inside the parenthesis. Likewise, at the end of the Figure 5 caption on page 11. (There is some typesetting problem there too.)

p. 13. 6 lines above Methods section. “Fig. 6 shows ...” rather than “Fig. 6 we show...”

p. 15. line 12. “there is less clear sign of an 8p-8h...” rather than “there is less clear signs of a 8p-8h...”

p. 18ff. The hyperlinks are missing from many of the references. At a quick look all references but the book of Ring and Schuck should have a hyperlink.

Version 1:

Reviewer comments:

Reviewer #1

(Remarks to the Author)

The authors addressed my remaining comment satisfactorily. I recommend the publication of the revised manuscript in Nature Communications.

Reviewer #3

(Remarks to the Author)

The authors have considered and carefully responded to the remarks in my previous reports. I believe that the manuscript has improved as a result of the feedback from the referees and I am happy to recommend it for publication in Nature Communications.

An observation: the work is relatively light on experimental data but heavy on theoretical interpretation. This situation is a result of the extreme difficulty of the experiments and the requirement of access to state-of-the art instrumentation. I hope that the publication of this paper will help motivate program advisory committees to approve beam time for more extensive studies of nuclei in this intriguing region of the nuclear landscape. I find the theoretical interpretation developed here quite compelling, but it needs to be tested by future experiments.

Open Access This Peer Review File is licensed under a Creative Commons Attribution 4.0 International License, which permits use, sharing, adaptation, distribution and reproduction in any medium or format, as long as you give appropriate credit to the original author(s) and the source, provide a link to the Creative Commons license, and indicate if changes were

made.

Answers to the reviewers' comments:

To reviewer #1

The authors addressed satisfactorily most of my comments. I find this work interesting and important enough to be published in Nature Communications.

Prior to recommending the publication, I ask the authors to further refine their discussion of the results obtained with the Vlow-k N3LO 2N interaction. An argument is made that the origin of the structural change between the two Mo isotopes are due the three-nucleon forces that drive the properties of the monopole part of the shell-model Hamiltonian. It is clear from Ref. [34], where the importance of 3N forces for the monopole part of the Hamiltonian was revealed, that effect comes from the genuine 3N force, i.e., the chiral 3N force or the force that originates from the Delta or other resonance excitations of nucleons. The discussed calculations with the Vlowk-softened N3LO two-nucleon interaction neglect not only the genuine 3N force but also the induced 3N interaction due to the Vlowk softening and due to the valence space truncation. I appreciate that the authors attempted a VS-IMSRG calculation and recognize that those are challenging in the present context. However, at the minimum, it should be acknowledged that the Vlowk N3LO 2N calculations are illustrative only and do not prove a genuine 3N force effect.

We thank the reviewer for the comments and for finding the work interesting and important enough to be published in Nature Communications.

We have incorporated an additional, elucidatory statement on page 12, immediately preceding Figure 5. This serves to explicitly acknowledge and emphasize the illustrative nature of our two-nucleon (2N) force calculations. We concur with the reviewer's assessment regarding the limitations of the Vlow-k N3LO 2N interaction calculations in definitively proving genuine three-nucleon (3N) force effects. Our intention in presenting these calculations was to provide a comparative framework and to illustrate potential trends, rather than to assert conclusive evidence of 3N force influences. The added statement clarifies this point. We believe this addition enhances the transparency of our methodology and strengthens the overall discussion of our results.

To reviewer #2

Title of this manuscript, ' Isospin-Symmetric Island of Inversion' is not fit to the contents of your manuscript. If you wish to describe this mass region as the "Island of Inversion" , you need to include a discussion of the nuclear structure of Sr, Zr, and Mo isotopes. If you intend to prepare a separate theoretical manuscript addressing this point, that version could be submitted to Nature Communications.

We appreciate the reviewer's comment regarding the title of our manuscript. We acknowledge that the current title "Isospin-Symmetric Island of Inversion" may not fully reflect the content of our work as presented. We modified the title to be more specific: "Abrupt Structural Transition in Exotic Molybdenum Isotopes unveils an Isospin-Symmetric Island of Inversion"

Your experimental results for Mo-84 and Mo-86 are very important for nuclear physics. Therefore, I recommend that you submit this draft to another journal, such as Physical Review C, Physics Letters B, or a similar venue, as soon as possible, since one year has already passed since the initial submission.

We sincerely thank the reviewer for the time and effort dedicated to evaluating our manuscript. We appreciate the acknowledgment of our work and the recognition of the importance of our experimental results on ⁸⁴Mo and ⁸⁶Mo for nuclear physics. Indeed, for this reason, we believe that the novelty and relevance of our results meet this journal's standards.

To reviewer #3

I believe that this work is worthy of publication in Nature Communications. The authors have responded appropriately to my previous report. However, in reading the revised manuscript, now some 12 months later, I have a few more comments for consideration and some minor corrections to note.

My first concern is one of terminology, in line with my first remark previously about the term “Island of Inversion”. The authors have already made changes in line with my previous remarks, but I find that in the last 12 months I have become more sensitive to statements like “magic numbers disappear”, and what the phrase “evolution of shell structure” really means. These terms are used in lines 10 and 11 of the Introduction. In Islands of Inversion the shell structure does not really disappear: it is still there but it is obscured by correlations that lower the energy of many-particle many-hole deformed configurations that dominate in the ground and low-lying states. The authors take this view in the remainder of the Introduction and indeed it is a theme of the paper. From this perspective it is more accurate to say “magic numbers seem to disappear”.

We want to thank the reviewer for defining our work worthy of publication in Nature Communications. We also thank the reviewer for the feedback regarding the terminology used in describing nuclear structure phenomena. It has been carefully considered and incorporated into the revised manuscript. The revised passage now reads: ‘magic numbers are obscured by correlations in nuclei with a large neutron excess and collective behaviour develops, associated with deformed nuclear shapes.’ This revision aims to convey that the shell structure is not disappearing, but rather being obscured by correlations, particularly in neutron-rich nuclei. It also emphasizes the development of collective behavior and its association with deformed nuclear shapes, which is a central theme of the paper. We believe that this change addresses the reviewer’s concern while maintaining scientific accuracy. It is hoped that this modification meets with the reviewer’s approval and more precisely captures the physics being discussed in the manuscript

I also recommend consideration of the phrase “evolution of shell structure” (line 11). As I see it, “evolution of nuclear structure” has a clear meaning in terms of what is observed. “Evolution of shell structure”, however, can be confusing because it must be inferred indirectly from what is observed and it depends on both single-particle energies and correlations, which are interrelated, as is evident in the body of this paper. In operational terms, the single particle energies depend on the basis space and on the interactions, which must be tuned to the basis space. An Island of Inversion is a region where the basis space must be opened up to include highly correlated np-nh excitations across the shell gap, which appear at lower energy than those in the normal (single major shell) basis space. This view accords with the theme of the paper. It is not obvious that this is a “consequence of the evolution of shell structure”. It is rather shell structure plus correlations. I strongly recommend revising the use of these phrases in lines 10 and 11 of the Introduction or otherwise making it explicit what the authors mean by them, particularly for a journal with an intended broader audience.

We agree that the phrase ‘evolution of shell structure’ can be ambiguous and potentially misleading, especially for a broader audience. In the revised version, we have rewritten line 10-11 of the introduction to be more concise and have removed this phrase. This change should make the introduction clearer and more accessible to readers from various backgrounds.

Now to some more specific comments and suggestions:

p. 4 end of 1st paragraph. Is the 2 e5 particles per second for the 86Mo or is this the total rate for all isotopes in the secondary beam? I guess the 86Mo rate but please clarify.

The rate shown in the text is the total rate for all of the isotopes. We appreciate the reviewer’s comment and modified the sentence “... approximately 2×10^5 particles per second” to “... approximately 2×10^5 particles per second in total”.

p.4 in Figure 1 caption. I suggest giving the actual v/c , say for the ^{86}Mo scattering case as an example, rather than just stating “higher” and “lower” velocity.

In the case for ^{86}Mo , the actual v/c was estimated to be 0.365 and 0.324 before and after passing the degrader. We explicitly wrote the value for ^{86}Mo in the caption of Figure 1.

p. 5. Fig. 2b. Can you comment on what the features might be at about 590 keV and 680 keV in the gamma-ray spectrum, which seem to be above background? What could be the impact of the feature at 590 keV be on the fast-peak intensity, and to what extent might it affect the deduced lifetime?

While it is normal and expected for some bins to deviate from the simulated value by more than one standard deviation and we expect this to be particularly noticeable in this case due to the low statistics in the spectrum, we conducted a comprehensive review of all count anomalies and we are confident that all reasonably observable transitions were taken into account in our analysis.

To reliably identify new transitions that could affect the lifetime measurements, we used the in-beam measurement taken with only the target mounted, without the degrader foil. This spectrum, shown in Figure 1 for comparison to the Figure 2 of the manuscript, has only the “fast” components of the peaks and is much clearer than the one with the degrader used for the lifetime measurement.

The target-only spectrum shows the $2^+ \rightarrow 0^+$ and $4^+ \rightarrow 2^+$ transitions. For the two regions at 590 keV and 680 keV there seem no known peaks from ^{86}Mo nor from laboratory-frame background to be included in the analysis. As we don’t observe any other transitions, we don’t expect a significant impact on the lifetime measurements.

Figure 1: The target-only spectrum of $^{9}\text{Be}(^{86}\text{Mo}, ^{86}\text{Mo}\gamma)$.

p. 6. Last line of first paragraph regarding the impact of the short 4^+ lifetime could perhaps be better phrased something along the following lines: “... which implies a short lifetime for the first 4^+ state and hence a limited impact on the 2^+ lifetime estimation, as included in the evaluation of systematic errors described below.”

We appreciate the suggestion from the reviewer. We agree that the suggested expression would be more clear to the readers. We replaced the part with the suggested one.

p. 6. Last line of Results section. This sentence, which has been changed in the revision process, is hard to follow. I don’t know what is meant. I thought the previous wording was clear (but maybe it was not accurate?). Please amend.

We estimated the uncertainty following the procedure described in detail in our previous response. In that explanation, we outlined the main assumptions we made and the steps we took, so that our method of evaluation can be transparent, clearly understood and considered reliable.

During the revision process we modified our original statement. Initially we wrote: "Such contribution is estimated using standard methods as described in [24]". We then changed this to: "Such contribution is estimated by comparison to the previous measurement [28]". We appreciate the reviewer's insightful comment, which helped us clarify this point. To enhance clarity, we have now further revised this sentence to read: "Such contribution is estimated based on the previous measurement [28]".

p. 8. Please add references to the various theoretical methods listed, specifically for QPSU3 and PHF. DNO-SM has references already given. The SM is probably well enough known that it doesn't require a reference. The PHF, however, appears to be a little unusual in that it considers a deformed HF state of specified np-nh structure, projects out 0+ and 2+ states and thus evaluates the B(E2). Is there a reference with detail on the procedure used? This would be useful for those who wish to understand a little more of the technicality behind these calculations.

We sincerely appreciate your comments regarding the theoretical methods used in our work. We agree with all your suggestions and implemented the changes in our revised manuscript. We added appropriate references for both the QPSU3 and PHF methods as requested. Regarding the PHF method, we understand your point. We provided a detailed reference that explains the specific procedure used, including the consideration of a deformed HF state with specified np-nh structure, the projection of 0+ and 2+ states, and the evaluation of B(E2).

Some minor corrections:

p. 2. Line 6 of Introduction. Insert "a" in "nuclei with a magic number. . ."

We corrected the grammatical mistake following the reviewer's suggestion.

p. 5. Fig. 2a and 2b captions. "The energy of the . . . transitions are. . ." should be "The energies of the . . . transitions are. . ."

We thank the reviewer for it. The expression is corrected with the suggestion.

p. 6. 2nd last line of 3rd paragraph. Should be "feeding ratios" not "feeding ratio".

We thank the reviewer for it. We corrected the expression.

p. 6. 4th paragraph line 2. "consists of a least" rather than "consists on a least"

We thank the reviewer for it. We corrected the expression.

p. 6. 5th paragraph line 1. "lifetimes of the first . . . were determined" rather than "lifetime of the first . . . were determined"

We thank the reviewer for it. The expression is corrected in accordance with the suggestion.

p. 7. About the middle of the text. Perhaps add "symmetries" to read "correspond to pseudo-SU(3) and quasi-SU(3) symmetries, respectively."

Following the reviewer's comment, we inserted the word "symmetries" to the sentence.

p. 9. 2 lines under Table 1. The full stop should be inside the parenthesis. Likewise, at the end of the Figure 5 caption on page 11. (There is some typesetting problem there too.)

We corrected the grammatical mistakes and the incorrect typesetting.

p. 13. 6 lines above Methods section. "Fig. 6 shows . . ." rather than "Fig. 6 we show. . ."

The expression was modified as suggested.

p. 15. line 12. "there is less clear sign of an 8p-8h. . ." rather than "there is less clear signs of a 8p-8h. . ."

We corrected the expression as suggested.

p. 18ff. The hyperlinks are missing from many of the references. At a quick look all references but the book of Ring and Schuck should have a hyperlink.

According to the Nature Communications guideline, the References has the following format: Author

list. Title of paper in sentence case. Name of journal **volume number**, initial-final page numbers or article number (year). Since there was no space for the hyperlink we removed them.